# Selective nitrogen insertion into aryl alkanes

Zheng Zhang[1], Qi Li[1], Zengrui Cheng [2], Ning Jiao [2] ✉ & Chun Zhang [1] ✉

Molecular structure-editing through nitrogen insertion offers more efficient and ingenious pathways for the synthesis of nitrogen-containing compounds, which could benefit the development of synthetic chemistry, pharmaceutical research, and materials science. Substituted amines, especially nitrogen-containing alkyl heterocyclic compounds, are widely found in nature products and drugs. Generally, accessing these compounds requires multiple steps, which could result in low efficiency. In this work, a molecular editing strategy is used to realize the synthesis of nitrogen-containing compounds using aryl alkanes as starting materials. Using derivatives of *O*-tosylhydroxylamine as the nitrogen source, this method enables precise nitrogen insertion into the C$sp^2$-C$sp^3$ bond of aryl alkanes. Notably, further synthetic applications demonstrate that this method could be used to prepare bioactive molecules with good efficiency and modify the molecular skeleton of drugs. Furthermore, a plausible reaction mechanism involving the transformation of carbocation and imine intermediates has been proposed based on the results of control experiments.

Amines are important organic compounds widely used in organic synthesis, materials science, and pharmaceutical research[1,2]. It is worth noting that some substituted amines, especially nitrogen-containing alkyl heterocyclic compounds, are key fragments of a variety of biologically active molecules (Fig. 1a)[3–8]. Furthermore, installing nitrogen-containing fragments into bioactive molecules is a key strategy for new drug development[9,10]. For example, converting the carbonyl group in erythromycin into amino fragments can produce azithromycin, an antibiotic with a wider antibacterial spectrum and better drug metabolism (Fig. 1b)[11]. However, inserting a nitrogen-containing unit into a carbon framework requires multiple steps with low efficiency, which is usually a challenging task (Fig. 1c)[12,13]. The synthetic strategy of molecular structure-editing could precisely add, swap, or delete single atom in the molecular skeleton, revolutionizing chemical synthesis route design[14–30]. The pioneering elegant achievements regarding nitrogen atom insertion into hydrocarbons have well illustrated that aromatic N-containing compounds can be intelligently prepared from easily available starting materials[31–37]. Herein, we developed a transition metal-free selective nitrogen insertion into aryl alkanes (Fig. 1d). This chemistry could be used to

obtain bioactive molecules with good efficiency and modify the molecular skeleton of drugs.

Aryl amine motifs are prevalent in various valuable compounds, such as drugs, natural products, and functional materials[38,39]. Therefore, developing practical methods to prepare derivatives of aryl amine is of general interest to the community of synthetic chemistry, drug development, materials science, and various chemical industries[40,41]. Generally, the methods to construct such important fragments include transition-metal-catalyzed cross-coupling reactions[42,43], reductive reactions of azo or nitro compounds[44], and direct arene aminations[45–50]. In recent decades, C-C bond-breaking transformations have been developed as powerful tool to construct new chemical bonds[51–54]. However, these methods usually convert the methylene unit into a byproduct, which could break the alkane ring (or chain) of the starting material[55–59]. Consequently, they cannot be used in the synthesis of nitrogen-containing heterocyclic products. In this present work, by leveraging a molecular editing strategy, we developed a method for the precise nitrogen insertion into the C$sp^2$-C$sp^3$ bond of aryl alkanes, which could access substituted amines, especially nitrogen-containing alkyl heterocyclic compounds, with good atom economy (Fig. 1d).

[1]Department of Chemistry, Institute of Molecular Plus, Tianjin Key Laboratory of Molecular Optoelectronic Science, School of Pharmaceutical Science and Technology, Tianjin University, Tianjin, China. [2]State Key Laboratory of Natural and Biomimetic Drugs, Chemical Biology Center, School of Pharmaceutical Sciences, Peking University, Beijing, China. ✉e-mail: jiaoning@pku.edu.cn; chunzhang@tju.edu.cn

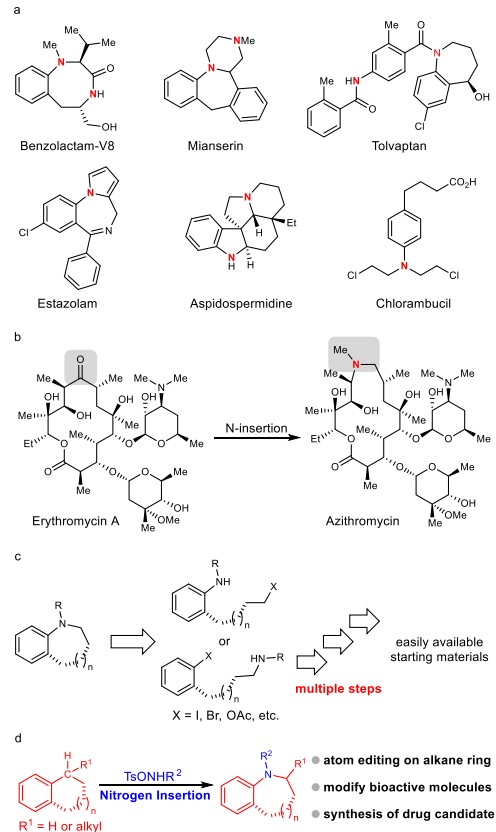

**Fig. 1 | The research background of selective nitrogen insertion into aryl alkanes. a** Representative bioactive molecules. **b** Development of azithromycin. **c** Normal retrosynthetic analysis of N-containing heterocyclic compounds. **d** This work: nitrogen atom insertion to extend the alkane ring (or chain).

## Results

### Optimization of the selective nitrogen insertion

The aim of our research is to develop more efficient and applicable synthesis methods for activating carbon-carbon bonds to form carbon-nitrogen bonds[60–63]. Based on our previous results regarding C-C bond cleavage amination[60–63], this study commenced with the one-pot and two-steps reaction of compounds **1** and **2** (Table 1). Firstly, the impact of different oxidants on the outcome of the metal-free reaction was investigated (Table 1, entries 1–4). The reaction with DDQ, 1,4-benzoquinone, or chloranil yielded product **3**, but using $H_2O_2$ as the oxidant did not result in the desired product. Among these oxidants, DDQ was the most effective (Table 1, entry 1). Interestingly, further studies suggested that $NaBH_4$ and HBpin could be used as reluctant at the second step of the reaction to access **3**, albeit with lower yields (Table 1, entries 5 and 6). Next, the reaction was conducted in different solvents (Table 1, entries 7–11). Using 2,2,2-trifluoroethanol as the solvent afforded **3** with a 95% yield (Table 1, entry 7). Furthermore, the desired product could be generated with lower yields when using $^i$PrOH, THF, or DCM as solvents (Table 1, entries 8–10). However, the reaction did not proceed when DMSO was used as the solvent (Table 1, entry 11). Moreover, increasing the temperature to 60 °C resulted in a slight decrease in the yield of **3** (Table 1, entry 12). The reaction proceeded at 0 °C, but only produced **3** with a 57% yield (Table 1, entry 13). Further control experiments illustrated that $H_2O$ is an important additive for improving the yield (Table 1, entry 14). We speculated that $H_2O$ plays a role in adjusting the solubility of the substances and additives. Importantly, without DDQ or $NaBH_3CN$, the present reaction did not work (Table 1, entries 15 and 16). These results suggest that these two reagents are essential for the present reaction.

### The investigation of substrate scope

After establishing the optimal reaction conditions, the tolerance of amination reagents was investigated (Fig. 2, **3–17**). Generally,

## Table 1 | The effects of different reaction conditions

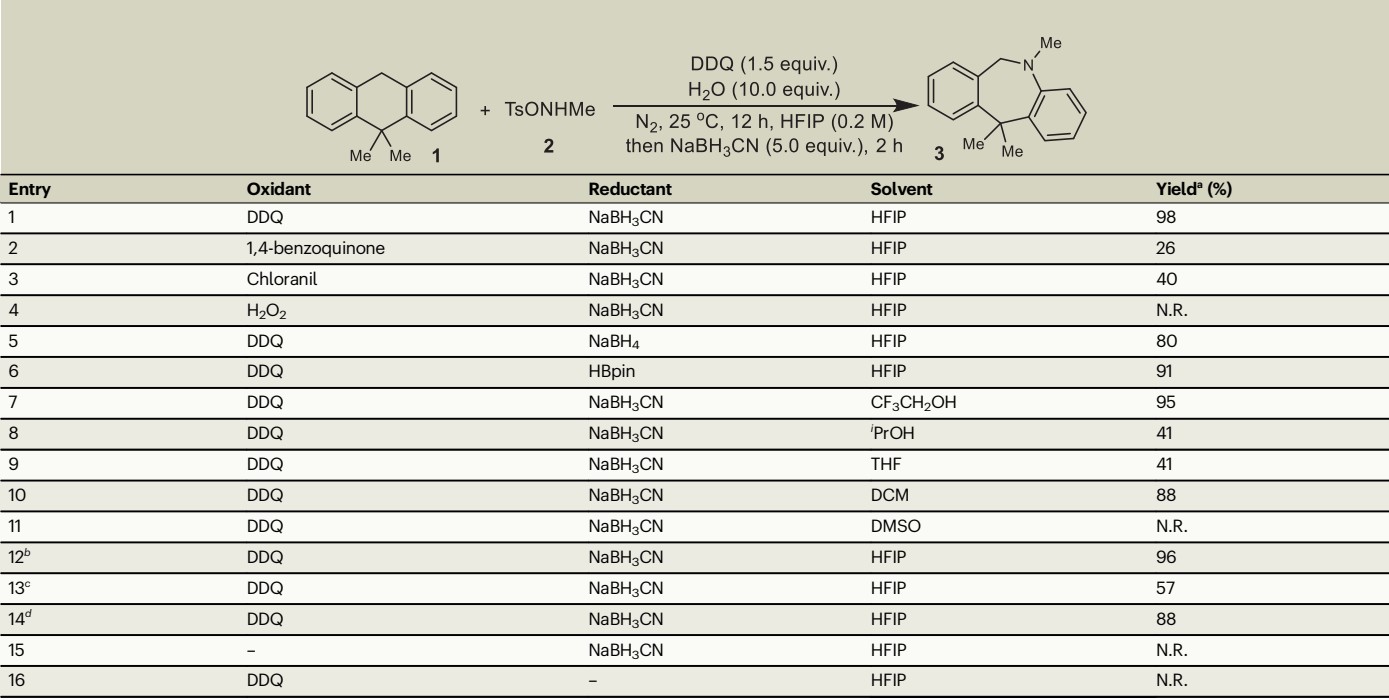

| Entry | Oxidant | Reductant | Solvent | Yield[a] (%) |
|---|---|---|---|---|
| 1 | DDQ | NaBH₃CN | HFIP | 98 |
| 2 | 1,4-benzoquinone | NaBH₃CN | HFIP | 26 |
| 3 | Chloranil | NaBH₃CN | HFIP | 40 |
| 4 | H₂O₂ | NaBH₃CN | HFIP | N.R. |
| 5 | DDQ | NaBH₄ | HFIP | 80 |
| 6 | DDQ | HBpin | HFIP | 91 |
| 7 | DDQ | NaBH₃CN | CF₃CH₂OH | 95 |
| 8 | DDQ | NaBH₃CN | ⁱPrOH | 41 |
| 9 | DDQ | NaBH₃CN | THF | 41 |
| 10 | DDQ | NaBH₃CN | DCM | 88 |
| 11 | DDQ | NaBH₃CN | DMSO | N.R. |
| 12[b] | DDQ | NaBH₃CN | HFIP | 96 |
| 13[c] | DDQ | NaBH₃CN | HFIP | 57 |
| 14[d] | DDQ | NaBH₃CN | HFIP | 88 |
| 15 | – | NaBH₃CN | HFIP | N.R. |
| 16 | DDQ | – | HFIP | N.R. |

[a]General reaction conditions: **1** (0.3 mmol), **2** (0.45 mmol), DDQ (0.45 mmol), H₂O (3.0 mmol), HFIP (1.5 mL), N₂, 25 °C, 12 h, then NaBH₃CN (1.5 mmol), 2 h. Isolated yield.
[b]60 °C instead of 25 °C.
[c]0 °C instead of 25 °C.
[d]No water.

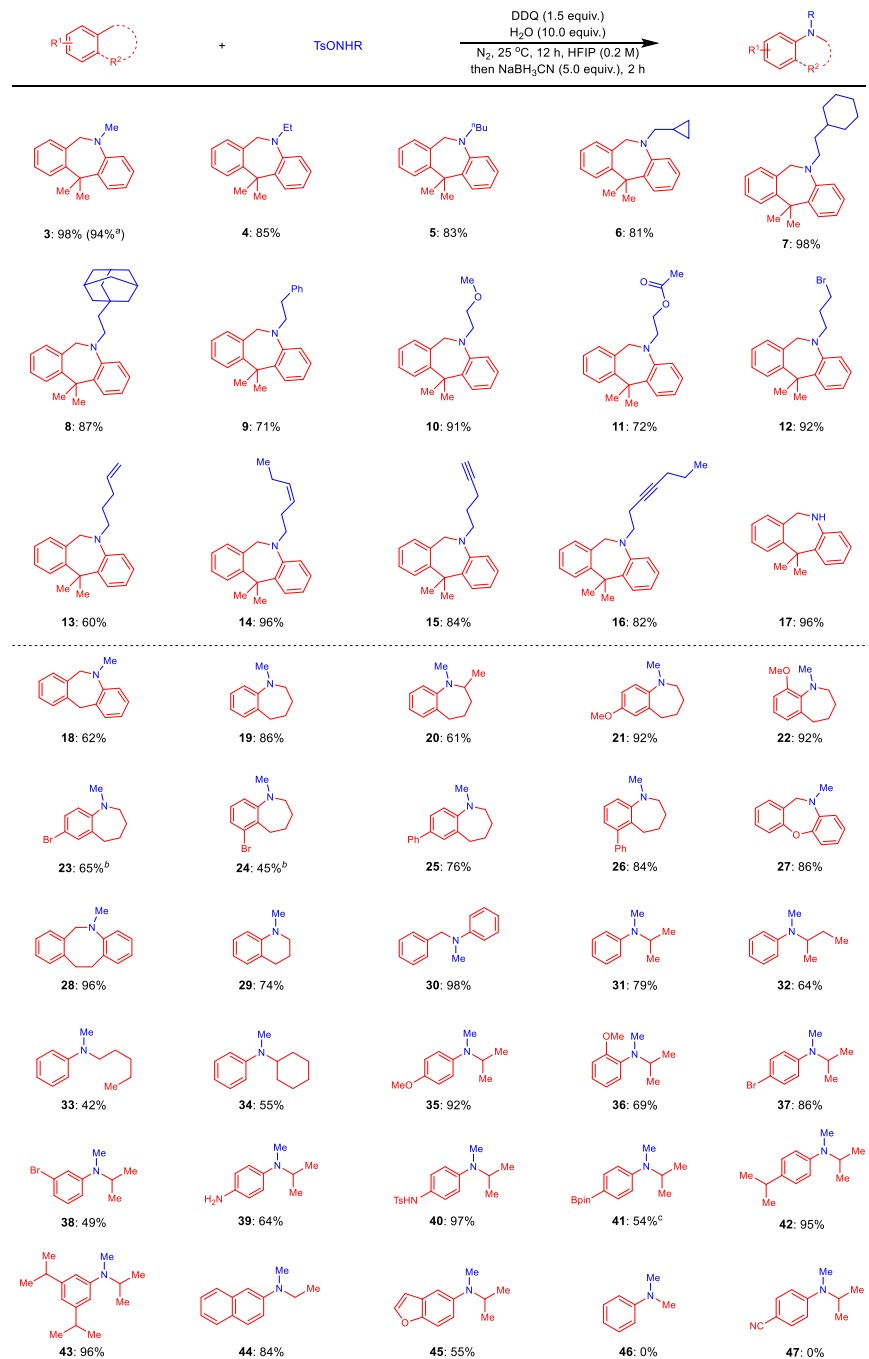

**Fig. 2 | Substrate scope of selective nitrogen insertion.** Aryl alkanes (0.3 mmol), amination reagent TsONHR (0.45 mmol), DDQ (0.45 mmol), H₂O (3.0 mmol), HFIP (1.5 mL), N₂, 25 °C, 12 h, then NaBH₃CN (1.5 mmol), 2 h, isolated yield. $^a$Aryl alkanes (5.0 mmol), amination reagent TsONHR (7.5 mmol), DDQ (7.5 mmol), H₂O (50.0 mmol), HFIP (25.0 mL), N₂, 25 °C, 12 h, then NaBH₃CN (25.0 mmol), 2 h. $^b$60 °C, 24 h instead of 25 °C, 12 h, then NaBH₃CN (25.0 mmol), 2 h. $^c$60 h instead of 12 h, then NaBH₃CN (25.0 mmol), 2 h.

substrates with various functional groups could be transformed into the desired product. First, the gram-scale reaction could work well with almost no decrease in the yield of the desired product (Fig. 2, **3**). When the methyl group was extended into a longer alkyl chain, the corresponding products could be obtained with a good yield (Fig. 2, **3**–**5**). Interestingly, amination reagents with secondary, tertiary, or benzyl carbon-hydrogen bonds could be used to synthesize the desired product (Fig. 2, **6** to **9**). Furthermore, some valuable but active groups, such as ether, ester, and bromine groups, could be smoothly transformed into the desired product from the starting materials (Fig. 2, **10** to **12**). Notably, both alkenyl and alkynyl groups

could be compatible with this amination reaction. Moreover, both terminal and internal carbon-carbon unsaturated bonds on the precursors could be transferred into the products with good to excellent yield (Fig. 2, **13** to **16**). Importantly, in addition to the tertiary amine products mentioned above, the present method could be used to insert secondary amines into the aryl alkanes (Fig. 2, **17**). The amination reagents with various valuable functional groups proved to be well compatible with the present chemistry (Fig. 2, **10** to **17**), unlike in our previous studies involving amination reagents, where they were ineffective[63]. Then, the substrate scope regarding aryl alkanes was studied (Fig. 2, **18**–**45**). Our present nitrogen insertion reaction could be

used as a powerful tool to synthesize nitrogen-containing alkyl het-erocyclic rings, which are key units in various bioactive molecules (Fig. 2, **18** to **29**). First, besides the bibenzyl position (Fig. 2, **3** and **18**), substrates with normal benzyl groups could be smoothly converted into the desired product (Fig. 2, **19** to **26**). Further studies suggested that functional groups, such as methoxy, bromo and phenyl groups, on the benzene ring could be tolerated (Fig. 2, **21–26**). Interestingly, in all of asymmetric examples, the reported products could be generated without producing any isomeric byproduct. For products **21, 23** and **25**, the -OMe, -Br, and -Ph groups at the para-position could facilitate the generation of benzyl carbocation, leading to the para-nitrogen insertion product. Conversely, the -Br and -Ph groups at the ortho-position would shield the benzyl, favoring meta-product formation (Fig. 2, **24** and **26**). The ortho-OMe group could promote the generation of ortho-benzyl carbocation without covering the benzyl site, allowing for the formation of product **22** as the ortho-insertion product. To our delight, oxygen-heterocyclic starting material could afford nitrogen insertion product with 86% yield (Fig. 2, **27**). Importantly, the present method could be used to construct 8-membered and 6-membered nitrogen-heterocyclic rings from readily available starting materials (Fig. 2, **28** and **29**). We next investigated the scope of noncyclic aryl alkanes under the optimal reaction conditions (Fig. 2, **30–45**). The results demon-strated both secondary and tertiary benzyl alkanes could be compatible with the nitrogen insertion reaction. Further scoping illustrated that valuable functional groups on the phenyl ring, such as -OMe, -Br, -NH2, -NHTs and -Bpin, could be transferred into the desired products (Fig. 2, **35–41**). Interestingly, when double or triple -iPr groups were installed on the phenyl ring, the present reaction could produce mono-amination products with great yield (Fig. 2, **42** and **43**). In addition to phenyl alkanes, other types of aryl alkanes, such as 2-ethylnaphthalene and 5-isopropylbenzofuran, could be efficiently converted into desired products (Fig. 2, **44** and **45**). Unfortunately, toluene and p-isopropyl benzonitrile cannot be converted into target products (Fig. 2, **46** and **47**).

## The studies of synthetic application

As the insertion of nitrogen atoms into biologically active molecules is an important strategy in drug development, the modification of various bioactive molecules using our present transformation has been investigated herein. Interestingly, nitrogen could be smoothly inserted into the derivatives of ibuprofen and dehydroabietic acid (Fig. 3a, **48** and **49**). The results of compound **48** suggest that the

benzyl C-H bond with an electron-withdrawing group is unfavorable, while the data from compound **49** illustrate that in this example the tertiary benzyl C-H bond on the carbon chain is much more favorable than secondary benzyl C-H bond on the carbon ring motif. Impor-tantly, this nitrogen insertion reaction could be used to prepare nonpeptidergic inhibitors of the human cytomegalovirus-encoded chemokine receptor in a practical manner (Fig. 3b). Under the opti-mal reaction conditions, the compound **50** could be converted into the key intermediate **51** with a 70% yield, and further substitution reaction with **52** could afford the desired product **54**. Compared with the previous method, our synthetic strategy uses cheaper starting materials, avoids inconvenient operations such as employing butyl lithium and microwave under high temperature, and provide higher yields with fewer operation steps[64,65]. The results in Fig. 3 illustrate the great application prospects of our developed synthetic methodology.

## Reaction mechanism study

To investigate the reaction mechanism, some control experi-ments have been designed (Fig. 4). Firstly, the source of hydrogen atoms in the product was investigated. Interestingly, when deut-erated starting material (**55-D1**) or deuterated water was used, the non-deuterated product could be isolated with a good yield (Fig. 4a, **1** and **2**). Importantly, if NaBD4 was employed, the deuterated product (D% = 95%) could be isolated with a 90% yield (Fig. 4a, **3**). These data could well demonstrate that the new hydrogen atom in the product originated from the reductant. Further kinetic isotope studies afforded a KIE value of 5.3:1, which supports the idea that C-H bond breaking is the selectivity-determining step (Fig. 4b). Moreover, a crossover experiment using a mixture of **57** and **55** as starting materials afforded **30** and **35** as products without the generation of **58** and **31** (Fig. 4c). These results imply that the reaction mechanism of Csp2-Csp3 bond cleavage and nitrogen insertion is an intramolecular reac-tion. To further investigate the effect of different substituents, competition experiments involving various phenyl alkanes were designed (Fig. 4d). Firstly, the reaction using an equal mixture of pentylbenzene and cyclohexylbenzene yielded 23% N-cyclohexyl-N-methylaniline with trace amounts of N-methyl-N-pentylaniline, suggesting that tertiary C-H bonds are more favorable than sec-ondary C-H bonds in the present reaction. Secondly, the reaction using an equal mixture of isopropyl benzene and 1-isopropyl-4-methoxybenzene afforded 92% of N-isopropyl-4-methoxy-N-methylaniline with trace N-isopropyl-N-methylaniline, demon-strating that the presence of electron-donating substituents in the aromatic ring could enhance the reaction. Then, TEMPO was used to trap the radical intermediate in the reaction system. However, the data illustrated that TEMPO did not affect the reaction, which afforded 83% of **3** and recovered 87% TEMPO (Fig. 4e). This data suggested that the present transformation does not proceed through a radical reaction process. Based on the above studies, the possible reaction mechanism has been proposed (Fig. 4f). First, carbocation **62** could be generated through the oxidation reaction between starting material **1** and DDQ or a nitrogen radical, which is produced from the reaction of DDQ and TsONHMe (Please see the Supplementary Figs 46–48 for more details)[61,62]. Then, the reaction of the animation reagent and **62** could afford the key intermediate **63**[57,66]. The following rearrangement could produce the imine inter-mediate **64**[67]. Then, further reduction could afford compound **3** as the final product.

## Discussion

In conclusion, we have developed a reaction for realizing nitrogen insertion into aryl alkanes. By employing a molecular editing strategy,

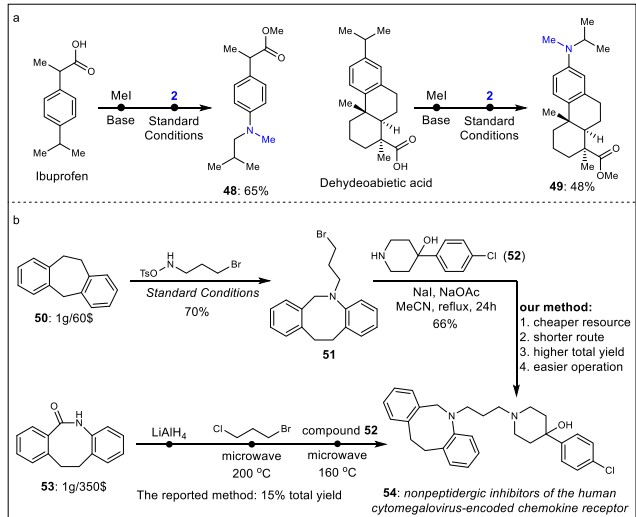

**Fig. 3 | Large scale reaction and further transformation of product. a** Modify various bioactive molecules. **b** Synthesis of bioactive molecule.

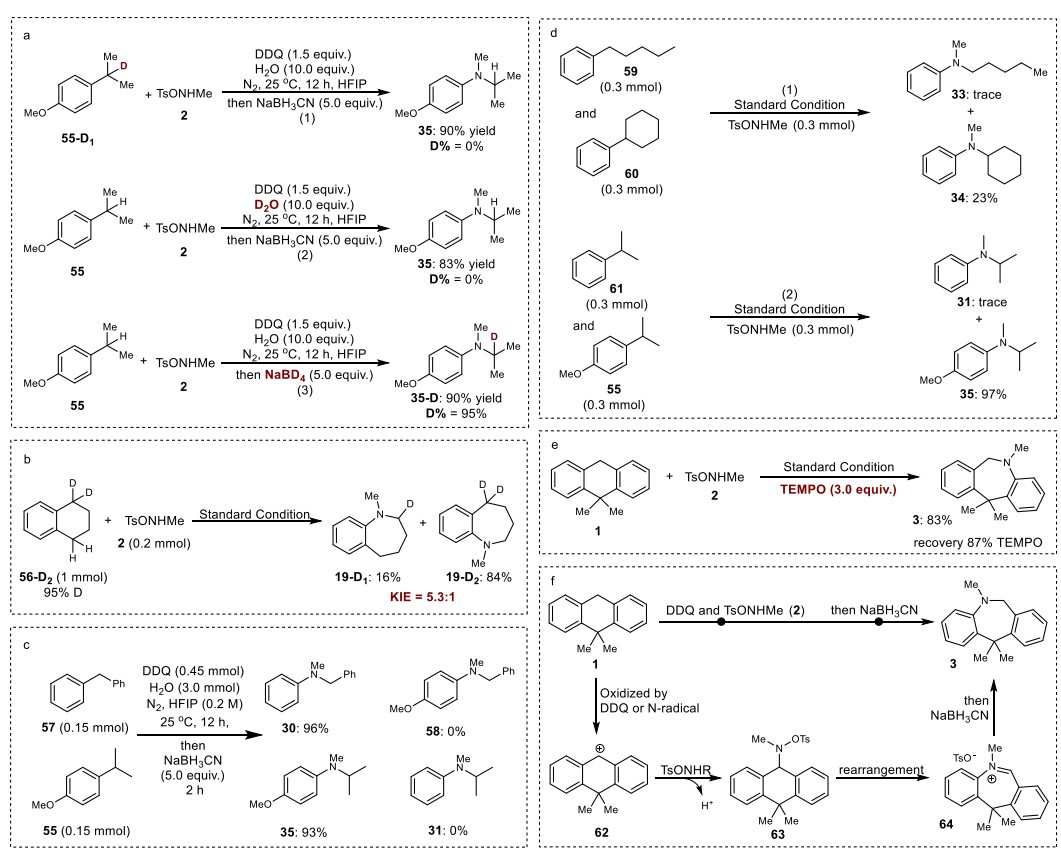

**Fig. 4 | The control experiments for reaction mechanism studies. a** Deuterium labeling experiments. **b** Kinetic isotope studies. **c** Crossover experiment. **d** Competition experiments of different kind phenyl alkanes. **e** Radical trapping experiment. **f** The proposed reaction mechanism.

this chemistry can expand the ring or lengthen the carbon chain under ambient reaction conditions. Notably, this chemistry, which is compatible with various functional groups, has been successfully used to modify drug molecules and synthesize bioactive compounds. Furthermore, the possible reaction mechanism has been proposed based on the results of radical trapping, deuterium labeling and isotope competition experiments. Further studies for synthetic applications are ongoing in our laboratory.

## Methods
In a 25 mL Schlenk tube that had been oven-dried and contained a stirring bar, the aminating reagent (0.45 mmol, 1.5 equiv.) and DDQ (0.45 mmol, 1.5 equiv.) were charged. The tube was then evacuated and back-filled under $N_2$ flow (this sequence was repeated three times). Water (3.0 mmol, 10.0 equiv.), aryl alkanes (0.3 mmol, 1.0 equiv.), and HFIP (1.5 mL) were added, then the mixture was stirred at room temperature for 12 h. Subsequently, $NaBH_3CN$ (1.5 mmol, 5.0 equiv.) was added to the reaction mixture and stirred for 2 h at room temperature. The reaction was quenched with 2.0 mL saturated $NaHCO_3$ aq. and 3.0 mL $H_2O$. The mixture was then extracted with DCM (3.0 mL × 3), and the combined organic layers were dried over $Na_2SO_4$, filtered, and concentrated by rotary evaporation. The residue was purified by silica gel chromatography (EtOAc/petroleum ether) to afford the desired product.

## Data availability
Supplementary information and chemical compound information accompany this paper at www.nature.com/ncomms. The data supporting the results of this work are included in this paper or in the Supplementary Information and are also available upon request from the corresponding author.

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

## Acknowledgements

We acknowledge the National Key R&D Program of China (No.2021YFA1501700), the NSFC (No. 22371203, 22131002, 22071005, and 22161142019), the Beijing Nova Program (No. Z201100006820099), and the Tencent Foundation for financial support.

## Author contributions

N. Jiao and C. Zhang conceived the research. Z. Zhang and Q. Li carried out experiments. Z. Zhang, Z. Cheng, N. Jiao, and C. Zhang analyzed results. N. Jiao and C. Zhang wrote the manuscript with input from all the authors.

## Competing interests

The authors declare no competing interests.
