## [Peer Review File · Nature Communications]

Selective Nitrogen Insertion into Aryl AlkanesEditorial Note: Parts of this Peer Review File have been redacted as indicated to remove third-party material where no permission to publish could be obtained.

REVIEWER COMMENTS

Reviewer #1 (Remarks to the Author):

The manuscript presents an innovative approach towards the synthesis of phenyl amines, particularly focusing on the selective nitrogen insertion into phenyl alkanes through a transition metal-free methodology. This work is poised to make a significant contribution to the fields of synthetic chemistry, drug development, given its potential to streamline the production of nitrogen-containing bioactive molecules. The emphasis on skeleton-editing as a strategy to enhance chemical synthesis is both timely and impactful, reflecting a deep understanding of the challenges and opportunities in modern chemical research.

However, while the paper is ambitious and presents a novel method that could indeed revolutionize aspects of synthetic chemistry, there are several areas where clarification, revision, or additional information would greatly enhance the manuscript's impact and scientific rigor.

Specific Comments:

1. The manuscript claims a broad applicability of the method to aryl alkanes, yet the examples provided seem to be limited to phenyl alkanes. This discrepancy between the claim and the demonstrated scope may inadvertently mislead readers about the versatility of the proposed method. The authors are advised to either clarify the limitations of their method with respect to the substrate scope or provide additional data demonstrating its applicability to a wider range of aryl alkanes.
2. The manuscript should explicitly state and emphasize that the nitrogen insertion process into aryl alkanes is a two-step reaction. This clarification is vital for readers to fully understand the experimental setup and the procedural nuances of the methodology being proposed.
3. The authors discuss their novel method without adequately connecting it to their previous research outputs, particularly a relevant study published in *Angewandte Chemie International Edition* (63, DOI: 10.1002/anie.202401318, 2024). Given the similarities between the conditions employed in the current manuscript and those in the referenced study, a thorough discussion comparing the two approaches would not only contextualize the present work but also highlight its novel contributions more effectively. This comparison could address advancements, limitations, or any modifications made to the previously published method.
4. The manuscript would benefit from a more comprehensive analysis of potential competing reactions, especially considering different alkyl groups (e.g., methyl, ethyl, isopropyl.....) and alkyl groups with varying electronic properties. Such an analysis is crucial for understanding the method's selectivity and efficiency.
5. Figure 1d appears to misrepresent the general structure of the starting materials used in the proposed synthetic method. Specifically, it is noted that the benzylic position should

contain one hydrogen atom to accurately reflect the compounds being discussed. This oversight may lead to confusion regarding the specificity and applicability of the described method.

The manuscript presents an innovative approach with significant potential implications for synthetic chemistry and related fields. Addressing the above comments, including the explicit clarification of the reaction steps, a thorough discussion of previous works, an in-depth analysis of competing reactions, and clarification on the scope regarding aromatic alkanes, will undoubtedly strengthen the paper. The authors are encouraged to make these revisions carefully to ensure that the manuscript accurately reflects the scope, novelty, and applicability of the research. I look forward to the revised manuscript, which promises to contribute significantly to the field.

Reviewer #2 (Remarks to the Author):

- Summary:

Significant Advance? Moderate.

Quality & Clarity Moderate.

Conclusions Supported? Yes.

SI Document? Good.

- Recommendation: Publish elsewhere, as a full paper, after revisions.

- Main Comments: Zhang and co-workers report the development of a synthetic method for the formal insertion of an alkylnitrene into the C(sp²)-C(sp³) bond of arylalkanes, yielding the corresponding tertiary aniline. This is an appealing synthetic transformation due to the importance of substituted anilines as synthetic targets, and because it offers a disconnection that follows the logic of the newly popularized “molecular editing” approaches. Impressive applications to the preparation of natural product derivatives (a.k.a. “unnatural products”) and pharmaceuticals were also reported.

The synthetic transformation reported consists in four distinct stages that are combined in a one-pot two-step operation: i) a benzylic oxidation yields a stabilized carbocation or possibly related species in equilibrium with the reactive cation (e.g., benzylic alcohol or HFIP ether); ii) nucleophilic attack of the carbocation by a nitrenoid species, here a N-alkylhydroxylamine O-tosylate; iii) a Schmidt-type or aza-Hock rearrangement (elsewhere instead described as a 1,2-Stieglitz shift; see *Org. Lett.* 2023, 25, 5795 and references therein) yielding an iminium ion intermediate and, finally iv) the reduction of the iminium ion using a borohydride.

Conceptually, the overall transformation is a logical, though rather modest, extension of previous reports including that of the authors' laboratories (cf. refs. 62-64, in addition to *ACS Catal.* 2019, 9, 2063 and *AAAS Research* 2020, 7947029), and that of the Hashmi (ref. 67) and Falck laboratories (*Chem. Sci.* 2022, 13, 4821, which should be cited). Indeed, stages i-iii were demonstrated in ref. 62 using DDQ as the oxidant and NaN₃ as the nitrenoid species to yield primary anilines following hydrolysis of the incipient iminium, alongside a single example combining stages i-iv including iminium reduction to afford a secondary aniline (compound 35 in ref. 62). *ACS Catal.* 2019, 9, 2063 combined stages i-iii using an electrochemical oxidation, an alkyl azide as the nitrenoid, and hydrolysis of the iminium intermediate to prepare secondary alkyl anilines, whereas *AAAS Research* 2020, 7947029

relied on the protonation of a styrene in the presence of RN₃ rather than electrochemical oxidation to access the same intermediates and products. Hashmi and colleagues (ref. 67) have instead combined stages ii–iii for most examples in their report using hydroxylamine O-sulfonates as the nitrenoid species, alongside a single example combining stages ii–iv (yielding compound 13 in ref. 67) and another combining stages i–iii with hydrolysis of the incipient iminium (18 → 3a in ref. 67). Finally, the Falck lab report started at the benzylic alcohol (or HFIP ether), but included examples combining stages ii–iv using HOSA as the nitrenoid reagent.

In sum, each of the stages of the reported nitrogen insertion transformation had clear precedents – and even the entire sequence i–iv in one example in ref 62 – dampening our enthusiasm for the novelty of the reported synthetic method.

This otherwise nice work certainly deserves publication. However, in view of the precedents above, doing so as a communication in *Nat. Commun.* seems inappropriate. Our recommendation to the editors is for this paper to instead be published elsewhere as a full paper further discussing key aspects of the transformation (see comments below). As *Nat. Commun.* does not publish full papers, another venue for publication will be more suitable, such as *Chem. Sci.* or *JACS*, for example.

1) Comparison of alkylarene N-insertion methods. A revised version of this manuscript as a full paper should include a frank critical discussion of the advantages, disadvantages and limitations of different reported methods (see above). An introductory scheme to graphically compare these methods may be advantageous as well.

For example, a key difference between this work and that reported earlier in ref. 62 (DDQ/NaN₃) is the use of N-alkylhydroxylamine O-tosylates as the nitrenoid, which enables the direct preparation of tertiary dialkyl anilines in lieu of secondary alkyl anilines (DDQ/NaN₃). However, the N-alkylhydroxylamine O-tosylates must be prepared in two steps from TsONHBoc (e.g., Mitsunobu alkylation followed by Boc deprotection; ref 67 SI), may have limited stabilities for extended storage as for most other hydroxylamine O-sulfonates (e.g., ref 67, SI), and some safety liabilities as energetic reagents. It is difficult to imagine a use case scenario where this is advantageous over the subsequent alkylation or reductive amination of secondary alkyl anilines obtained by the earlier DDQ/NaN₃ method (ref. 62). One could imagine a “molecular editing” campaign to obtain diverse tertiary dialkyl anilines from a single alkyl arene, but this would require facile access to varied shelf-stable N-alkylhydroxylamine O-tosylates. Given the above, it seems unlikely that such building blocks will be readily available.

Has an approach using far more readily available alkyl azides (e.g., DDQ/RN₃/NaBH₃CN) been attempted, for the sake of comparison?

2) Limitations. The authors discuss the substrate scope of the reaction and compatible functional groups (e.g., Fig. 2). Unfortunately, there is no discussion of the limitations of the nitrogen insertion reaction. A revised version of this manuscript as a full paper should include a discussion of substrates that failed to undergo the desired reaction and/or resulted in undesirable side reactions. For example, were some functional groups tested and found not to be tolerated? Were direct electrophilic amination products predominant for some substrates (cf. 18 → 19 in ref. 67; *Adv. Synth. Catal.* 2021, 363, 2783)? [An attempt with estrone methyl ether or a related substrate would be instructive in this regard.] What are the upper and lower limits of the electronic properties of alkyl arenes that can undergo the

nitrogen insertion reaction? Are highly electron rich or highly electron poor alkyl arenes compatible? Are alkylated heteroaromatic substrates also competent in the reaction (alkylpyridines, alkyl(benzo)thiophenes, alkyl(benzo)furans, etc.)? Can the nitrogen insertion reaction take place at an Ar-CH₃ bond (no examples were shown)? Some of these limitations may have been mentioned in the earlier DDQ/NaN₃ paper (ref. 62), but it would be worthwhile to restate them herein in a full discussion of the limitations of the synthetic method.

3) Selectivity. Some substrates raise interesting chemo- or regioselectivity issues that should be discussed in greater detail in revised version of this manuscript as a full paper. After all, the title of this manuscript emphasizes the selectivity of the N-insertion reaction, but there is hardly any discussion of the relevant parameters that influence its selectivity.

For example, were the regiomeric N-insertion isomers detected in the preparation of 23, 24, 42 (ibuprofen) or 43 (dehydroabiatic acid), and if so in what ratio? How is the regioselectivity observed in polyalkylated arenes dependent on the position and electronic properties of other substituents? That appears to be the case for some tetralin derivatives (e.g. ortho-insertion in 22 vs. meta-insertion in 24), but a discussion of these effects is missing. One would assume that the DDQ oxidation is the selectivity-determining step, but were there examples where a synthetically useful selectivity (e.g. $\geq 5:1$) could not be achieved?

• Other Comments / Suggestions / Corrections:

- Abstract: In our view the abstract reads like an introduction paragraph, and hardly provides a satisfactory summary of the synthetic method and results (reagents employed, nature of the reaction/mechanism, etc.). We believe this could be improved.

- Introduction, line 6: "Azithromycin" => "azithromycin" (does not require capitalization).

- The authors do not seem to have evaluated TFA as a co-solvent (e.g., Table 1). Given some of its advantages (low cost compared to HFIP), and the fact that TFA was identified as the best solvent for the closely related transformation reported by the Hashmi lab (18 => 3a in ref. 67), it may be worthwhile to screen it as well and report the results.

- Page 5 and Fig 3a: "dehydroabiatic acid" => "dehydroabiatic acid" [typos]

- Figure 4 and the accompanying discussion in the text: "cross reaction experiment" => "crossover experiment".

- Although we believe that the proposed mechanism is sound and most likely correct, we wonder whether the radical trapping experiment with TEMPO (Fig. 4d) is sufficiently reliable to rigorously rule out any radical involvement or intermediate. To do so, one would need to verify that DDQ does react with TEMPO (i.e. is oxidation to the oxoammonium ion possible under these conditions? If so, is the oxoammonium also competent in the benzylic oxidation?). Moreover, if a benzylic radical was generated, it could be trapped by TEMPO leading to a benzylic hydroxylamine ether under ionization equilibrium with the benzylic carbocation. An additional control experiment could also be carried out to establish that no oxidation (i.e. H-atom abstraction) of the N-alkylhydroxylamine O-tosylates by DDQ may take place.

- The mechanistic experiments, such as the intramolecular deuterium labeling experiment in

Fig. 4b, suggest that DDQ oxidation is the selectivity-determining step. While it is correct to note the KIE, we believe that the statement: "... supports the idea that C-H bond breaking is the kinetic determining step" is improperly phrased. Nothing in this experiment established that the benzylic oxidation is slower than its trapping by the N-alkylhydroxylamine O-tosylates or else the subsequent aza-Hock rearrangement; it merely establishes that the benzylic oxidation must precede these steps. Consequently, it would be more accurate to write "supports the idea that C-H bond breaking is the selectivity-determining step" instead.

- Could the incipient iminium ion be trapped with nucleophiles other than a hydride? That might be possible, though we will concede that such a discussion is better left to a separate publication.

- Ref. 14 was published in 2023, not 2003 (typo).

- Ref. 38 is an odd choice. The paper is about the Cryo-EM structure of a protein. Yes it was crystallized with a bound ligand (donitriptan) that does include an alkylniline, but it is hardly the main point of the cited paper...

Reviewer #3 (Remarks to the Author):

In this study, Zhang, Jiao, and their team have innovatively devised a protocol for nitrogen insertion into a C-C bond. Their approach hinges on the generation of a carbocation intermediate, which subsequently transforms into an amine intermediate and undergoes aryl migration. This series of rearrangements ultimately yields the desired product. Notably, the substrate scope encompasses intriguing compounds that would be challenging to synthesize using alternative methods. The reaction conditions and mechanisms are grounded in their previous work published in *Nature Chemistry* (volume 11, pages 71-77, 2019), differing primarily in the post-treatment approach. I firmly believe that replacing sodium azide with ammonia reagent is crucial, often ensuring superior efficiency in certain reactions. Given its novelty and practical utility, this reaction deserves publication in *Nature Communications*. However, a minor revision, coupled with a few additional experiments, is recommended prior to its publication.

1. The reviewer noticed that the manuscript included some asymmetric examples (21-24), but still achieved good regioselectivity. Please analyze the influence of different types of substituents on regioselectivity. In this process, some typical electron-withdrawing substituents, such as cyano, ester, and trifluoromethyl groups, as well as some electrically neutral substituents, such as aryl and alkyl groups, should be included and discussed.

2. Including some additional experiments can enrich the content of the article. For example, by adding an oxidant during the post-treatment process of substrate 27, a quinoline product can be obtained. The author can further expand the application of this reaction.

3. Additional referencing to N-atom insertion into C-C bond, should be included. *Org. Lett.* 2023, 25, 5795–5799

REVIEWER COMMENTS

Reviewer #1 (Remarks to the Author):

The manuscript presents an innovative approach towards the synthesis of phenyl amines, particularly focusing on the selective nitrogen insertion into phenyl alkanes through a transition metal-free methodology. This work is poised to make a significant contribution to the fields of synthetic chemistry, drug development, given its potential to streamline the production of nitrogen-containing bioactive molecules. The emphasis on skeleton-editing as a strategy to enhance chemical synthesis is both timely and impactful, reflecting a deep understanding of the challenges and opportunities in modern chemical research.

However, while the paper is ambitious and presents a novel method that could indeed revolutionize aspects of synthetic chemistry, there are several areas where clarification, revision, or additional information would greatly enhance the manuscript's impact and scientific rigor.

Response: Thank you for the kind and positive comments. This paper has been carefully improved base on the following constructive suggestions. Thanks again!

Specific Comments:

1. The manuscript claims a broad applicability of the method to aryl alkanes, yet the examples provided seem to be limited to phenyl alkanes. This discrepancy between the claim and the demonstrated scope may inadvertently mislead readers about the versatility of the proposed method. The authors are advised to either clarify the limitations of their method with respect to the substrate scope or provide additional data demonstrating its applicability to a wider range of aryl alkanes.

Response: Thank you for the kind suggestion. The present chemistry could afford the desired

product with good yield when 2-ethylnaphthalene or 5-isopropylbenzofuran was used as the starting material (see eq. 1-1 and eq. 1-2). These data have been added into the main text. Unfortunately, the reaction with 5-butylbenzo[b]thiophene or 5,6,7,8-tetrahydroquinoline did not work (see eq. 1-3 and eq. 1-4). These data have been added into the SI. The above results demonstrate that besides phenyl alkanes, other kinds of aryl alkanes could work with the present chemistry. Thanks again!

2. The manuscript should explicitly state and emphasize that the nitrogen insertion process into aryl alkanes is a two-step reaction. This clarification is vital for readers to fully understand the experimental setup and the procedural nuances of the methodology being proposed.

Response: Thank you for the kind suggestion. We have revised the first sentence in the part of RESULTS AND DISCUSSION, and explicitly stated this chemistry is a two-steps reaction: “Based on our previous results regarding C-C bond cleavage amination,⁶¹⁻⁶⁴ this study commenced with the one-pot and two-steps reaction of compounds **1** and **2** (Table 1).” Furthermore, we have revised the sentence: “Interestingly, further studies suggested that NaBH₄ and HBpin could be used as reluctant to access **3**, albeit with lower yields (Table 1, entries 5 and 6).” into “Interestingly, further studies suggested that NaBH₄ and HBpin could be used as reluctant at the second step of the reaction to access **3**, albeit with lower yields (Table 1, entries 5 and 6).” Thanks again!

3. The authors discuss their novel method without adequately connecting it to their previous research outputs, particularly a relevant study published in *Angewandte Chemie International Edition* (63, DOI: 10.1002/anie.202401318, 2024). Given the similarities between the conditions employed in the current manuscript and those in the referenced study, a thorough discussion comparing the two approaches would not only contextualize the present work but also highlight its novel contributions more effectively. This comparison could address advancements, limitations, or any modifications made to the previously published method.

Response: Thank you for the kind suggestion. Based our previous research (cited references 61-64 in text), this chemistry has been designed and optimized. In comparison with the work of ref. 63,

we added 10 equivalents of H₂O as an additive to improve the yield of the desired product. Additionally, we carefully investigated the reaction time to shorten the reductive step's duration. Importantly, the substrate scope of amination reagents is much broader than in our previous work. For example, amination reagents with alkenyl, alkynyl, ether, ester, and bromine groups work well in the present chemistry, whereas they did not work in our previous studies (ref 63). This information has been added into the text. Moreover, we conducted a competition experiment between the present nitrogen-insertion reaction and the previous ring-opening amination reaction. Interestingly, the result yielded the product of the present chemistry as the main product without generating any ring-opening product (see eq. 1-5 and eq. 1-6). This data has been added into the SI. Thanks again!

4. The manuscript would benefit from a more comprehensive analysis of potential competing reactions, especially considering different alkyl groups (e.g., methyl, ethyl, isopropyl.....) and alkyl groups with varying electronic properties. Such an analysis is crucial for understanding the method's selectivity and efficiency.

Response: Thank you for the kind suggestion. Firstly, the reaction using an equal mixture of pentylbenzene and cyclohexylbenzene could produce 23% N-cyclohexyl-N-methylaniline with trace amounts of N-methyl-N-pentylaniline (see eq. 1-7). This data suggests that secondary C-H bonds are much more favorable than primary C-H bonds in the present reaction. Secondly, the reaction using an equal mixture of isopropyl benzene and 1-isopropyl-4-methoxybenzene could afford 92% of N-isopropyl-4-methoxy-N-methylaniline with trace amounts of N-isopropyl-N-methylaniline (see eq. 1-8). This result demonstrates that the presence of electron-donating substituents in the aromatic ring could benefit the reaction. These competition experiments could support that the idea that reaction mechanism involves the oxidation of C-H bonds to produce carbocation intermediates. The above information has been added into the text. Thanks again!

5. Figure 1d appears to misrepresent the general structure of the starting materials used in the proposed synthetic method. Specifically, it is noted that the benzylic position should contain one

hydrogen atom to accurately reflect the compounds being discussed. This oversight may lead to confusion regarding the specificity and applicability of the described method.

Response: Thank you for the kind suggestion. This error has been revised (see eq. 1-9). Thanks again!

The manuscript presents an innovative approach with significant potential implications for synthetic chemistry and related fields. Addressing the above comments, including the explicit clarification of the reaction steps, a thorough discussion of previous works, an in-depth analysis of competing reactions, and clarification on the scope regarding aromatic alkanes, will undoubtedly strengthen the paper. The authors are encouraged to make these revisions carefully to ensure that the manuscript accurately reflects the scope, novelty, and applicability of the research. I look forward to the revised manuscript, which promises to contribute significantly to the field.

Response: Thank you for the kind and positive comments. This paper has been carefully improved base on your constructively suggestions. All the questions and suggestions have been responded to point-to-point. Thanks again!

Reviewer #2 (Remarks to the Author):

• Summary:

Significant Advance? Moderate.

Quality & Clarity Moderate.

Conclusions Supported? Yes.

SI Document? Good.

• Recommendation: Publish elsewhere, as a full paper, after revisions.

• Main Comments: Zhang and co-workers report the development of a synthetic method for the formal insertion of an alkylnitrene into the C(sp²)-C(sp³) bond of arylalkanes, yielding the corresponding tertiary aniline. This is an appealing synthetic transformation due to the importance of substituted anilines as synthetic targets, and because it offers a disconnection that follows the logic of the newly popularized “molecular editing” approaches. Impressive applications to the preparation of natural product derivatives (a.k.a. “unnatural products”) and pharmaceuticals were also reported.

The synthetic transformation reported consists in four distinct stages that are combined in a one-pot two-step operation: i) a benzylic oxidation yields a stabilized carbocation or possibly related species in equilibrium with the reactive cation (e.g., benzylic alcohol or HFIP ether); ii)

nucleophilic attack of the carbocation by a nitrenoid species, here a N-alkylhydroxylamine O-tosylate; iii) a Schmidt-type or aza-Hock rearrangement (elsewhere instead described as a 1,2-Stieglitz shift; see *Org. Lett.* 2023, 25, 5795 and references therein) yielding an iminium ion intermediate and, finally iv) the reduction of the iminium ion using a borohydride.

Conceptually, the overall transformation is a logical, though rather modest, extension of previous reports including that of the authors' laboratories (cf. refs. 62-64, in addition to *ACS Catal.* 2019, 9, 2063 and *AAAS Research* 2020, 7947029), and that of the Hashmi (ref. 67) and Falck laboratories (*Chem. Sci.* 2022, 13, 4821, which should be cited). Indeed, stages i~iii were demonstrated in ref. 62 using DDQ as the oxidant and NaN₃ as the nitrenoid species to yield primary anilines following hydrolysis of the incipient iminium, alongside a single example combining stages i~iv including iminium reduction to afford a secondary aniline (compound 35 in ref. 62). *ACS Catal.* 2019, 9, 2063 combined stages i~iii using an electrochemical oxidation, an alkyl azide as the nitrenoid, and hydrolysis of the iminium intermediate to prepare secondary alkyl anilines, whereas *AAAS Research* 2020, 7947029 relied on the protonation of a styrene in the presence of RN₃ rather than electrochemical oxidation to access the same intermediates and products. Hashmi and colleagues (ref. 67) have instead combined stages ii~iii for most examples in their report using hydroxylamine O-sulfonates as the nitrenoid species, alongside a single example combining stages ii~iv (yielding compound 13 in ref. 67) and another combining stages i~iii with hydrolysis of the incipient iminium (18 → 3a in ref. 67). Finally, the Falck lab report started at the benzylic alcohol (or HFIP ether), but included examples combining stages ii~iv using HOSA as the nitrenoid reagent.

In sum, each of the stages of the reported nitrogen insertion transformation had clear precedents – and even the entire sequence i~iv in one example in ref 62 – dampening our enthusiasm for the novelty of the reported synthetic method.

This otherwise nice work certainly deserves publication. However, in view of the precedents above, doing so as a communication in *Nat. Commun.* seems inappropriate. Our recommendation to the editors is for this paper to instead be published elsewhere as a full paper further discussing key aspects of the transformation (see comments below). As *Nat. Commun.* does not publish full papers, another venue for publication will be more suitable, such as *Chem. Sci.* or *JACS*, for example.

Response: Thank you for reviewing this work. The present chemistry has been design to afford valuable aryl amine products from easily available starting materials using the novel “molecular editing” strategy. This paper has been carefully improved base on your constructive suggestions. Below, all the questions and suggestions have been responded to point-to-point. Furthermore, the key reference *Chem. Sci.*, 2022, 4821 has been cited as ref. 57, and the reference *Org. Lett.* 2023, 5795 has been added into the text at this time. Thanks again!

1) Comparison of alkylarene N-insertion methods. A revised version of this manuscript as a full paper should include a frank critical discussion of the advantages, disadvantages and limitations of different reported methods (see above). An introductory scheme to graphically compare these

methods may be advantageous as well.

For example, a key difference between this work and that reported earlier in ref. 62 (DDQ/NaN₃) is the use of N-alkylhydroxylamine O-tosylates as the nitrenoid, which enables the direct preparation of tertiary dialkyl anilines in lieu of secondary alkyl anilines (DDQ/NaN₃). However, the N-alkylhydroxylamine O-tosylates must be prepared in two steps from TsONHBoc (e.g., Mitsunobu alkylation followed by Boc deprotection; ref 67 SI), may have limited stabilities for extended storage as for most other hydroxylamine O-sulfonates (e.g., ref 67, SI), and some safety liabilities as energetic reagents. It is difficult to imagine a use case scenario where this is advantageous over the subsequent alkylation or reductive amination of secondary alkyl anilines obtained by the earlier DDQ/NaN₃ method (ref. 62). One could imagine a “molecular editing” campaign to obtain diverse tertiary dialkyl anilines from a single alkyl arene, but this would require facile access to varied shelf-stable N-alkylhydroxylamine O-tosylates. Given the above, it seems unlikely that such building blocks will be readily available.

Has an approach using far more readily available alkyl azides (e.g., DDQ/RN₃/NaBH₃CN) been attempted, for the sake of comparison?

Response: Thank you for your kind questions. The present chemistry utilizes TsONH₂ can afford the desired product with a 96% yield. However, employing NaN₃ as a nitrogen-source can only yields trace amounts of the product and generates numerous byproducts (see eq. 2-1 and eq. 2-2). TsONH₂ proves to be a superior choice in this chemistry. These screening data regarding nitrogen source have been added into the SI. Thanks again!

2) Limitations. The authors discuss the substrate scope of the reaction and compatible functional groups (e.g., Fig. 2). Unfortunately, there is no discussion of the limitations of the nitrogen insertion reaction. A revised version of this manuscript as a full paper should include a discussion of substrates that failed to undergo the desired reaction and/or resulted in undesirable side reactions. For example, were some functional groups tested and found not to be tolerated? Were direct electrophilic amination products predominant for some substrates (cf. 18 \rightarrow 19 in ref. 67; Adv. Synth. Catal. 2021, 363, 2783)? [An attempt with estrone methyl ether or a related substrate would be instructive in this regard.] What are the upper and lower limits of the electronic properties of alkyl arenes that can undergo the nitrogen insertion reaction? Are highly electron rich or highly electron poor alkyl arenes compatible? Are alkylated heteroaromatic substrates also

competent in the reaction (alkylpyridines, alkyl(benzo)thiophenes, alkyl(benzo)furans, etc.)? Can the nitrogen insertion reaction take place at an Ar-CH₃ bond (no examples were shown)? Some of these limitations may have been mentioned in the earlier DDQ/Na₃N paper (ref. 62), but it would be worthwhile to restate them herein in a full discussion of the limitations of the synthetic method.

Response: Thank you for your kind question and suggestion. The limitation of the nitrogen insertion reaction is evident in aryl alkanes bearing strong electron-withdrawing groups, such as -CO₂Me, -CN, and -CF₃, which did not yield the desired results (see table 2-1). These data has been added into SI.

Table 2-1. Failed reactions

In our present chemistry, direct electrophilic amination products were not detected in all of the examples. Following a valuable suggestion from the reviewer, the reaction using estrone methyl ether as the starting material yielded the desired product without any direct electrophilic amination byproducts. However, the product was mixed with some inseparable by-product, thus limiting us to reporting an NMR yield only (see eq. 2-3). This result has been added into SI.

The present chemistry yielded the desired product with good yield when 2-ethylnaphthalene or 5-isopropylbenzofuran was used as the starting material (see eq. 2-4 and eq. 2-5). These data have been added into the main text. Unfortunately, the reaction with 5-butylbenzo[b]thiophene, 5,6,7,8-tetrahydroquinoline, or toluene did not succeed (see eq. 2-6, eq. 2-7 and eq. 2-8). These data have been added into the SI. Thanks again!

3) Selectivity. Some substrates raise interesting chemo- or regioselectivity issues that should be discussed in greater detail in revised version of this manuscript as a full paper. After all, the title of this manuscript emphasizes the selectivity of the N-insertion reaction, but there is hardly any discussion of the relevant parameters that influence its selectivity.

For example, were the regiomer N-insertion isomers detected in the preparation of **23**, **24**, **42** (ibuprofen) or **43** (dehydroabietic acid), and if so in what ratio? How is the regioselectivity observed in polyalkylated arenes dependent on the position and electronic properties of other substituents? That appears to be the case for some tetralin derivatives (e.g. ortho-insertion in **22** vs. meta-insertion in **24**), but a discussion of these effects is missing. One would assume that the DDQ oxidation is the selectivity-determining step, but were there examples where a synthetically useful selectivity (e.g. $\geq 5:1$) could not be achieved?

Response: Thank you for your kind questions and suggestions. In all of these examples, the reported product could be produced without generating any isomeric byproducts. In the cases of products **23** and **24**, the presence of a -Br group at para-position could facilitate the generation of a benzyl carbocation, whereas a -Br group at ortho-position would shield the benzyl site, favoring the formation of the meta-product. In comparison, the -OMe could facilitate the generation of an ortho-benzyl carbocation and does not cover the benzyl site, thus affording product **22** as the ortho-insertion product. The result of **42** (new number is **46**) support the hypothesis that the carbocation is a key intermediate in the reaction mechanism, as benzyl C-H with electron-withdrawing groups are unfavorable. Furthermore, the data of **43** (new number is **47**) illustrate that the benzyl C-H bond on the carbon chain is much more favorable than on the carbon ring motif. These discussions have been added into the text. Thanks again!

• Other Comments / Suggestions / Corrections:

- Abstract: In our view the abstract reads like an introduction paragraph, and hardly provides a satisfactory summary of the synthetic method and results (reagents employed, nature of the reaction/mechanism, etc.). We believe this could be improved.

Response: Thank you for your kind suggestion. The abstract has been improved. Thanks again!

- Introduction, line 6: “Azithromycin” => “azithromycin” (does not require capitalization).

Response: Thank you for your kind suggestion. This error has been revised. Thanks again!

- The authors do not seem to have evaluated TFA as a co-solvent (e.g., Table 1). Given some of its advantages (low cost compared to HFIP), and the fact that TFA was identified as the best solvent for the closely related transformation reported by the Hashmi lab (18 => 3a in ref. 67), it may be worthwhile to screen it as well and report the results.

Response: Thank you for your kind suggestion. We screened the reaction conditions using TFA as a solvent or co-solvent (see Table 2-2). The results suggested that the reaction using HFIP was the most effective. These data have been added into the SI. Thanks again!

Table 2-2. The results using TFA as solvent or co-solvent

Entry	Solvent	Yield (%)
1	TFA	47
2	TFA : HFIP = 2:1	31
3	TFA : HFIP = 1:1	41
4	TFA : HFIP = 1:2	61
5	HFIP	98

- Page 5 and Fig 3a: “dehydroabiatic acid” => “dehydroabietic acid” [typos]

Response: Thank you for your kind suggestion. This error has been revised. Thanks again!

- Figure 4 and the accompanying discussion in the text: “cross reaction experiment” => “crossover experiment”.

Response: Thank you for your kind suggestion. This error has been revised. Thanks again!

- Although we believe that the proposed mechanism is sound and most likely correct, we wonder

whether the radical trapping experiment with TEMPO (Fig. 4d) is sufficiently reliable to rigorously rule out any radical involvement or intermediate. To do so, one would need to verify that DDQ does react with TEMPO (i.e. is oxidation to the oxoammonium ion possible under these conditions? If so, is the oxoammonium also competent in the benzylic oxidation?). Moreover, if a benzylic radical was generated, it could be trapped by TEMPO leading to a benzylic hydroxylamine ether under ionization equilibrium with the benzylic carbocation. An additional control experiment could also be carried out to establish that no oxidation (i.e. H-atom abstraction) of the N-alkylhydroxylamine O-tosylates by DDQ may take place.

Response: Thank you for your kind question and suggestion. Under the reaction conditions of the control experiments, there was no reaction observed between DDQ and TEMPO, as both compounds could be recovered equivalently (see eq. 2-9). Furthermore, the radical trapping experiment with TEMPO (Fig. 4d) yielded 83% of the desired product **3** with an 87% recovery rate of TEMPO (see eq. 2-10). This data supports that TEMPO did not participate in the reaction to produce product **3**. This data has been updated in the text.

In the absence of aryl alkanes, DDQ could degrade TsONHMe and generate numerous unknown by-products (see eq. 2-11). Further control experiments involving the addition of starting material **1** when TsONHMe was stirred with DDQ for 3 hours yielded the desired product with 92% yield (see eq. 2-12). Moreover, in the absence of TsONHMe, DDQ could oxidize the aryl alkane (see eq. 2-13). These results suggest that carbocation **60** could be generated through the oxidation reaction between starting material **1** and DDQ or a nitrogen radical, which is produced from the reaction of DDQ and TsONHMe. The relevant discussion in the text has been revised, and the data have been added into SI. Thanks again!

- The mechanistic experiments, such as the intramolecular deuterium labeling experiment in Fig. 4b, suggest that DDQ oxidation is the selectivity-determining step. While it is correct to note the KIE, we believe that the statement: "... supports the idea that C-H bond breaking is the kinetic determining step" is improperly phrased. Nothing in this experiment established that the benzylic oxidation is slower than its trapping by the N-alkylhydroxylamine O-tosylates or else the subsequent aza-Hock rearrangement; it merely establishes that the benzylic oxidation must precede these steps. Consequently, it would be more accurate to write "supports the idea that C-H bond breaking is the selectivity-determining step" instead.

Response: Thank you for your kind suggestion. This error has been revised. Thanks again!

- Could the incipient iminium ion be trapped with nucleophiles other than a hydride? That might be possible, though we will concede that such a discussion is better left to a separate publication.

Respond: Thank you for your kind question and suggestion. We attempted to use Grignard reagent or phenylboronic acid to trap the iminium ion, but none of these reactions yielded the desired product (see eq. 2-14, eq. 2-15 and eq. 2-16). These data have been added into SI. Thanks again!

- Ref. 14 was published in 2023, not 2003 (typo).

Response: Thank you for your kind suggestion. This error has been revised. Thanks again!

- Ref. 38 is an odd choice. The paper is about the Cryo-EM structure of a protein. Yes it was crystallized with a bound ligand (donitriptan) that does include an alkyylaniline, but it is hardly the main point of the cited paper...

Response: Thank you for your kind suggestion. This reference has been removed. Thanks again!

Reviewer #3 (Remarks to the Author):

In this study, Zhang, Jiao, and their team have innovatively devised a protocol for nitrogen

insertion into a C-C bond. Their approach hinges on the generation of a carbocation intermediate, which subsequently transforms into an amine intermediate and undergoes aryl migration. This series of rearrangements ultimately yields the desired product. Notably, the substrate scope encompasses intriguing compounds that would be challenging to synthesize using alternative methods. The reaction conditions and mechanisms are grounded in their previous work published in Nature Chemistry (volume 11, pages 71-77, 2019), differing primarily in the post-treatment approach. I firmly believe that replacing sodium azide with ammonia reagent is crucial, often ensuring superior efficiency in certain reactions. Given its novelty and practical utility, this reaction deserves publication in Nature Communications. However, a minor revision, coupled with a few additional experiments, is recommended prior to its publication.

Respond: Thank you for your kind and positive comments. This paper has been carefully improved base on following constructive suggestions. Thanks again!

1. The reviewer noticed that the manuscript included some asymmetric examples (21-24), but still achieved good regioselectivity. Please analyze the influence of different types of substituents on regioselectivity. In this process, some typical electron-withdrawing substituents, such as cyano, ester, and trifluoromethyl groups, as well as some electrically neutral substituents, such as aryl and alkyl groups, should be included and discussed.

Respond: Thank you for your kind question and suggestion. In all of the asymmetric examples, the reported products could be produced without generating any isomeric byproduct. For example, in the cases of products **21** and **23**, the presence of the -OMe or -Br group at para-position could facilitate the generation of a benzyl carbocation. However, the -Br group at the ortho-position (compound **24**) would shield the benzyl site, favoring the formation of the meta-product. In comparison, the ortho-OMe group could facilitate the generation of an ortho-benzyl carbocation and does not cover the benzyl site, thus affording product **22** as the ortho-insertion product. These discussions have been added into the text.

Under the optimal reaction conditions, aryl alkanes with strong electron-withdrawing groups, such as -CO₂Me, -CN, and -CF₃, did not yield desired results (see table 3-1). These data have been added into SI.

Table 3-1. Failed reactions

Aryl alkanes with alkyl groups smoothly transformed into the desired product (see compound **42**, **43**, **46** and **47** in the main text). Interestingly, starting materials with a phenyl group proved compatible with the optimal reaction conditions, yielding a single product without isomers (see eq. 3-1 and eq. 3-2). The phenyl group at the para-position could facilitate the generation of a benzyl

carbocation (eq. 3-1), whereas a phenyl group at the ortho-position would shield the benzyl site, favoring the formation of the meta-product (eq. 3-2). The above information have been added into the text. Thanks again!

2. Including some additional experiments can enrich the content of the article. For example, by adding an oxidant during the post-treatment process of substrate 27, a quinoline product can be obtained. The author can further expand the application of this reaction.

Respond: Thank you for your kind suggestion. Employing an oxidative reaction to replace the reductive step could convert the intermediate into quinolone product. As shown in table 3-2, the reaction with KMnO_4 successfully achieved the target reaction, yielding the quinolone with a 26% yield. These data have been added into the SI. Thanks again!

Table 3-2. One-pot reaction to synthesize quinoline

Entry	oxidation condition	Yield (%)
1	Pt/C (0.1 equiv.), air	0
2	H_2O_2 (30 wt% in H_2O , 5.0 equiv.)	0
3	KMnO_4 (5.0 equiv.)	26

3. Additional referencing to N-atom insertion into C-C bond, should be included. *Org. Lett.* 2023, 25, 5795–5799

Respond: Thank you for your kind suggestion. This literature has been added. Thanks again!

REVIEWER COMMENTS

Reviewer #1 (Remarks to the Author):

The authors have revised the manuscript and generally improved the quality by solving numerous raised issues.

I have no remaining scientific concerns.

Reviewer #2 (Remarks to the Author):

The authors have provided in this first revision a good-faith effort to address several of our comments and those of the other referees as well. In particular, we are satisfied with the revisions to address our earlier comments regarding the limitations of the reaction (Comment 2 from our earlier report) and its selectivity (Comment 3). The result is a significantly improved manuscript.

Overall, our judgement on the novelty of the chemistry presented in this manuscript has not fundamentally changed. Each of the stages of the reported nitrogen insertion transformation having had clear precedents, the work represents, to a degree, the repackaging of established reactivity under the currently more fashionable theme of “skeletal editing” transformations (notwithstanding the demonstrated synthetic utility of that transformation). Therefore, though this work is more than worthy of publication, its readers would be better deserved by presenting it as a full paper in a journal that publishes such articles (e.g., JACS, Chem. Sci.). However, we humbly concede, in view of the other referees’ comments, that ours might be a minority opinion.

Consequently, we now recommend that the editors of Nat. Commun. reconsider the manuscript after further revisions to provide the reader with a better comparison with previously reported methods for alkylarene N-insertion and mechanistically analogous (i.e., refs. 61 in particular, but also 57, 61, 63, 66 and 67).

For our full comments, see the submitted file.

Reviewer #3 (Remarks to the Author):

The author has well addressed the issues I raised in the revision. Therefore, I agree to accept this manuscript for publication.

• **Manuscript ID:** NCOMMS-24-14090A

• **Title:** Selective Nitrogen Insertion into Aryl Alkanes.

• **Author(s):** Zheng Zhang, Qi Li, Zengrui Cheng, Ning Jiao,* and Chun Zhang*

• **Summary:**

Significant Advance? Moderate.

Quality & Clarity Moderate.

Conclusions Supported? Yes.

SI Document? Good.

• **Recommendation:** Reconsider after revisions.

• **Main Comments:** The authors have provided in this first revision a good-faith effort to address several of our comments and those of the other referees as well. In particular, we are satisfied with the revisions to address our earlier comments regarding the limitations of the reaction (Comment 2 from our earlier report) and its selectivity (Comment 3). The result is a significantly improved manuscript.

Overall, our judgement on the novelty of the chemistry presented in this manuscript has not fundamentally changed. Each of the stages of the reported nitrogen insertion transformation having had clear precedents, the work represents, to a degree, the repackaging of established reactivity under the currently more fashionable theme of “skeletal editing” transformations (notwithstanding the demonstrated synthetic utility of that transformation). Therefore, though this work is more than worthy of publication, its readers would be better deserved by presenting it as a full paper in a journal that publishes such articles (e.g., *JACS*, *Chem. Sci.*). However, we humbly concede, in view of the other referees’ comments, that ours might be a minority opinion.

Consequently, we now recommend that the editors of *Nat. Commun.* reconsider the manuscript after further revisions to provide the reader with a better comparison with previously reported methods for alkylarene N-insertion and mechanistically analogous transformations (i.e., refs. 61 in particular, but also 57, 61, 63, 66 and 67).

For our full comments, see the submitted file.

As we previously indicated (Comment 1 of our earlier referee report), a revised version of this manuscript as a full paper should include a frank critical discussion of the advantages and disadvantages different reported methods. Even though all the reaction steps of the alkylarene N-insertion were previously established in these earlier references – not to mention the strongly related precedents in Boyer-Schmidt-Aube-type rearrangements involving benzylic azides/azidiums – there is no detailed mention of these in introduction of the manuscript that would allow the reader to compare and contrast this work with previous reports. The only contextualization provided by the authors rather abruptly appears in the first sentence of the results and discussion section: “Based on our previous

results regarding C–C bond cleavage amination,⁶⁰⁻⁶³ this study commenced...”. Such a discussion should be added, notably including for the reader a clear, face-to-face comparison of the different methods.

For example, the reaction below was shown in ref. 61 using DDQ/NaN₃:

[Redacted]

(Scheme above taken from *Nat. Chem.* **2019**, *11*, 71–77)

The reader is not provided with a straightforward comparison using the same substrate, under conditions either optimized for the DDQ/NaN₃ reaction (ref. 61), or those for the DDQ/TsONH₂ reaction (this manuscript). Unfortunately, “**1a**” above (4-isopropylbiphenyl) was not included in this manuscript, but alternately the optimized for the DDQ/NaN₃ reaction could be carried out with an equivalent substrate included therein for a proper face-to-face comparison (e.g., **53**, for example)

• N-insertion under conditions optimized for DDQ/NaN₃ (*Nat. Chem.* **2019**, *11*, 71-77):

[Redacted]

• N-insertion under conditions optimized for DDQ/TsONHR (NCOMMS-24-14090A):

There is no mention in ref. 61 of cyclic alkylarenes. In their rebuttal letter, the authors indicated that, for 9,9-dimethyl-9,10-dihydroanthracene as the substrate, only traces of the N-insertion reaction were obtained when the DDQ/NaN₃ reaction was conducted *under the conditions optimized for the DDQ/TsONH₂ reaction*.

(Scheme above taken from the NCOMMS-24-14090A rebuttal letter)

That is not the proper comparison for the benefit of the readership. The reaction with DDQ/NaN₃ should be conducted under its own optimized conditions to establish a proper comparison, e.g.,

- N-insertion under conditions optimized for DDQ/NaN₃ (*Nat. Chem.* **2019**, *11*, 71-77):

[Redacted]

- N-insertion under conditions optimized for DDQ/TsONHR (NCOMMS-24-14090A):

It is possible (even likely) that this will properly establish the superiority of the new DDQ/TsONH₂ insertion for 9,9-dimethyl-9,10-dihydroanthracene, but the readers must be provided with the proper evidence.

Moreover, it should be established whether the limitation of the DDQ/NaN₃ insertion reaction is limited to this specific cyclic substrate (9,9-dimethyl-9,10-dihydroanthracene), or whether it is applicable to other/most cyclic arylalkanes. Examples with simple substrates like indane or tetralin should be sufficient to provide evidence in this regard.

Finally, we previously raised the issue of the benefits of an early *N*-alkylation (claimed as an advantage in this manuscript) against that of a late *N*-alkylation reaction. Here again, providing a face-to-face comparison would provide the readership with a proper understanding of the pros and cons of the different methods. (A third comparison using an *alkyl* azide rather than NaN₃ may also be worthwhile – cf. *ACS Catal.* **2019**, *9*, 2063 and *AAAS Research* **2020**, 7947029.) A comparison substrate/target taken from this paper (e.g.,

31-34) would limit the amount of additional experimental for the authors to carry out while establishing a face-to-face comparison of these methods. For example:

- N-insertion under conditions optimized for DDQ/NaN₃ (*Nat. Chem.* **2019**, *11*, 71-77), followed with a late *N*-alkylation:

[Redacted]

- N-insertion under conditions optimized for DDQ/TsONHR (NCOMMS-24-14090A) with an early *N*-alkylation:

• Other (minor) Comments / Suggestions / Corrections:

- We believe it would be preferable for the benefit of the readers to include directly within Fig. 2 substrates that did not react, if space allows, rather than relegating them to the SI document (e.g., Fig. S7).

- In the discussion of the selectivity of the N-insertion for dehydroabietic acid (**47**), the authors say "...the benzyl C-H bond on the carbon chain is much more favorable than on the carbon ring motif." This is strictly correct for **47**, but could mislead some readers tempted to extrapolate this conclusion. In view of the mechanistic experiments of Fig. 4d, wouldn't it be better to say that the tertiary C-H bond was more reactive than the secondary C-H bond instead?

That said, if the authors have separate evidence that a cyclic 2° C-H is less reactive than an acyclic 2° C₂H (or for cyclic 3° C-H vs. acyclic 3° C-H), that would be useful data to add to Fig. 4.

- In the discussion of the mechanistic experiments comparing **57** and **58** (Fig.4d) giving **33** and **34**, respectively, the text says: "suggesting that secondary C-H bonds are more favorable than primary C-H bonds in the present reaction." Shouldn't that instead read "suggesting that tertiary C-H bonds are more favorable than secondary C-H bonds in the present reaction." ?

- As we indicated in our earlier comment, we do not require that the authors explore the trapping of the iminium with other nucleophiles. However, the attempts to add Grignard reagents directly to a reaction mixture containing an excess of aqueous HFIP (and presumably reduced DDQ) appear misguided at best (page S56). If the authors truly want to explore this reactivity in their future work, they would likely fare better by considering

acid-tolerant π -nucleophiles with a Mayr $N > 4$ knowb to add to iminiums. These could include allylsilanes/stannanes, 1-methylindole, silyl enol ethers, etc. (see *J. Am. Chem. Soc.* **2000**, *122*, 7226 and cited & citing papers, for instance).

- Abstract line 1: “**The** molecular...” => “Molecular...”
- Abstract, line 7: “derivates” => “derivatives”
- Abstract, line 8: “this **innovative** method” => “this method”
- Abstract, line 11: “synthetic application” => “synthetic applications”
- Introduction, line 9: “could precise swap” => “could precisely swap”
- Studies of synthetic applications, line 12: “...and **yields higher** with fewer operation steps.” => “and **provide higher yields** with fewer operation steps.”
- Reaction mechanism study, line 8: “a **cross-reaction** experiment” => “a **crossover** experiment”
- SI pages S40-41. These products are quinol**ines**, not quinol**ones**.

- **Manuscript ID:** NCOMMS-24-14090A
- **Title:** Selective Nitrogen Insertion into Aryl Alkanes.
- **Author(s):** Zheng Zhang, Qi Li, Zengrui Cheng, Ning Jiao,* and Chun Zhang*

- **Summary:**

Significant Advance?	Moderate.
Quality & Clarity	Moderate.
Conclusions Supported?	Yes.
SI Document?	Good.

- **Recommendation:** Reconsider after revisions.
- **Main Comments:** The authors have provided in this first revision a good-faith effort to address several of our comments and those of the other referees as well. In particular, we are satisfied with the revisions to address our earlier comments regarding the limitations of the reaction (Comment 2 from our earlier report) and its selectivity (Comment 3). The result is a significantly improved manuscript.

Overall, our judgement on the novelty of the chemistry presented in this manuscript has not fundamentally changed. Each of the stages of the reported nitrogen insertion transformation having had clear precedents, the work represents, to a degree, the repackaging of established reactivity under the currently more fashionable theme of “skeletal editing” transformations (notwithstanding the demonstrated synthetic utility of that transformation). Therefore, though this work is more than worthy of publication, its readers would be better deserved by presenting it as a full paper in a journal that publishes such articles (e.g., *JACS*, *Chem. Sci.*). However, we humbly concede, in view of the other referees’ comments, that ours might be a minority opinion.

Consequently, we now recommend that the editors of *Nat. Commun.* reconsider the manuscript after further revisions to provide the reader with a better comparison with previously reported methods for alkylarene N-insertion and mechanistically analogous transformations (i.e., refs. 61 in particular, but also 57, 61, 63, 66 and 67).

For our full comments, see the submitted file.

As we previously indicated (Comment 1 of our earlier referee report), a revised version of this manuscript as a full paper should include a frank critical discussion of the advantages and disadvantages different reported methods. Even though all the reaction steps of the alkylarene N-insertion were previously established in these earlier references – not to mention the strongly related precedents in Boyer-Schmidt-Aube-type rearrangements involving benzylic azides/azidiums – there is no detailed mention of these in introduction of the manuscript that would allow the reader to compare and contrast this work with

previous reports. The only contextualization provided by the authors rather abruptly appears in the first sentence of the results and discussion section: “Based on our previous results regarding C–C bond cleavage amination,⁶⁰⁻⁶³ this study commenced...”. Such a discussion should be added, notably including for the reader a clear, face-to-face comparison of the different methods.

For example, the reaction below was shown in ref. 61 using DDQ/NaN₃:

[Redacted]

(Scheme above taken from *Nat. Chem.* **2019**, *11*, 71–77)

The reader is not provided with a straightforward comparison using the same substrate, under conditions either optimized for the DDQ/NaN₃ reaction (ref. 61), or those for the DDQ/TsONH₂ reaction (this manuscript). Unfortunately, “**1a**” above (4-isopropylbiphenyl) was not included in this manuscript, but alternately the optimized for the DDQ/NaN₃ reaction could be carried out with an equivalent substrate included therein for a proper face-to-face comparison (e.g., **53**, for example)

• N-insertion under conditions optimized for DDQ/NaN₃ (*Nat. Chem.* **2019**, *11*, 71-77):

[Redacted]

• N-insertion under conditions optimized for DDQ/TsONHR (NCOMMS-24-14090A):

There is no mention in ref. 61 of cyclic alkylarenes. In their rebuttal letter, the authors indicated that, for 9,9-dimethyl-9,10-dihydroanthracene as the substrate, only traces of the N-insertion reaction were obtained when the DDQ/NaN₃ reaction was conducted *under the conditions optimized for the DDQ/TsONH₂ reaction*.

(Scheme above taken from the NCOMMS-24-14090A rebuttal letter)

That is not the proper comparison for the benefit of the readership. The reaction with DDQ/ NaN_3 should be conducted under its own optimized conditions to establish a proper comparison, e.g.,

- N-insertion under conditions optimized for DDQ/ NaN_3 (*Nat. Chem.* **2019**, *11*, 71-77):

[Redacted]

- N-insertion under conditions optimized for DDQ/ TsONHR (NCOMMS-24-14090A):

It is possible (even likely) that this will properly establish the superiority of the new DDQ/ TsONH_2 insertion for 9,9-dimethyl-9,10-dihydroanthracene, but the readers must be provided with the proper evidence.

Moreover, it should be established whether the limitation of the DDQ/ NaN_3 insertion reaction is limited to this specific cyclic substrate (9,9-dimethyl-9,10-dihydroanthracene), or whether it is applicable to other/most cyclic arylalkanes. Examples with simple substrates like indane or tetralin should be sufficient to provide evidence in this regard.

Finally, we previously raised the issue of the benefits of an early *N*-alkylation (claimed as an advantage in this manuscript) against that of a late *N*-alkylation reaction. Here again, providing a face-to-face comparison would provide the readership with a proper understanding of the pros and cons of the different methods. (A third comparison using an

alkyl azide rather than NaN_3 may also be worthwhile – cf. *ACS Catal.* **2019**, *9*, 2063 and *AAAS Research* **2020**, 7947029.) A comparison substrate/target taken from this paper (e.g., **31-34**) would limit the amount of additional experimental for the authors to carry out while establishing a face-to-face comparison of these methods. For example:

- N-insertion under conditions optimized for DDQ/ NaN_3 (*Nat. Chem.* **2019**, *11*, 71-77), followed with a late N-alkylation:

[Redacted]

- N-insertion under conditions optimized for DDQ/TsONHR (NCOMMS-24-14090A) with an early N-alkylation:

Response: Thank you for your kind question and suggestion. In order to better compare the differences between this method and previous methods, we conducted the following experiments:

1) The 1-isopropyl-4-methoxybenzene has been selected as the starting material to synthesize the desired product under our present reaction conditions and the conditions of previous reports (see below, eq.1 and eq. 2). The results suggest that our present chemistry would be the better choice as it afforded a highly yield. Furthermore, under the optimal reaction conditions of the reported method, using TsONHMe as nitrogen source did not work (see below, eq.3). This data has been added into the SI. Thank again!

Notably, using the reported method to prepare the cyclic products only afford trace amounts (Table 1). These results could well illustrate the advantage of our present method in synthesizing cyclic products. The above data has been added into the SI. Thank again!

Table 1. The preparation of cyclic product using the reported method

2) The desired product, N-isopropyl-4-methoxy-N-methylaniline, could be synthesized by the following two synthetic routes. The first route was designed based on the reported method, which afforded the desired product with a total yield of 21% (eq. 4). Meanwhile, the second synthetic route could produce the desired product with a total yield of 53%, using our present method as a critical synthesis step (eq. 5). These results have been added into the SI. Thanks again!

3) The reaction using alkyl azide as a nitrogen source only gives trace amount of the desired product, regardless of whether under our present conditions or reported methods (ref. 61) (eq. 6 and eq. 7). This data has been added into the SI. Thanks again!

Other (minor) Comments / Suggestions / Corrections:

- We believe it would be preferable for the benefit of the readers to include directly within Fig. 2 substrates that did not react, if space allows, rather than relegating them to the SI document (e.g., Fig. S7).

Response: Thank you for your kind suggestion. Representative examples about no reaction have been added into Fig. 2. Thanks again!

- In the discussion of the selectivity of the N-insertion for dehydroabietic acid (**47**), the authors say "...the benzyl C-H bond on the carbon chain is much more favorable than on the carbon ring motif." This is strictly correct for **47**, but could mislead some readers tempted to extrapolate this conclusion. In view of the mechanistic experiments of Fig. 4d, wouldn't it be better to say that the tertiary C-H bond was more reactive than the secondary C-H bond instead?

That said, if the authors have separate evidence that a cyclic 2° C-H is less reactive than an acyclic 2° C-H (or for cyclic 3° C-H vs. acyclic 3° C-H), that would be useful data to add to Fig. 4.

Response: Thank you for your kind suggestion. This error has been revised. Thanks again!

- In the discussion of the mechanistic experiments comparing **57** and **58** (Fig.4d) giving **33** and **34**, respectively, the text says: "suggesting that secondary C-H bonds are more favorable than primary C-H bonds in the present reaction." Shouldn't that instead read

“suggesting that tertiary C-H bonds are more favorable than secondary C-H bonds in the present reaction.” ?

Response: Thank you for your kind suggestion. This error has been revised. Thanks again!

- As we indicated in our earlier comment, we do not require that the authors explore the trapping of the iminium with other nucleophiles. However, the attempts to add Grignard reagents directly to a reaction mixture containing an excess of aqueous HFIP (and presumably reduced DDQ) appear misguided at best (page S56). If the authors truly want to explore this reactivity in their future work, they would likely fare better by considering acid-tolerant π -nucleophiles with a Mayr $N > 4$ knowb to add to iminiums. These could include allylsilanes/stannanes, 1-methylindole, silyl enol ethers, etc. (see *J. Am. Chem. Soc.* **2000**, *122*, 7226 and cited & citing papers, for instance).

Response: Thank you for your kind suggestion. The reaction using allylsilanes, 1-methylindole, and silyl enol ethers have been studied, but no desired product could be detected (Table 2). These data has been added into SI. Thanks again!

Table 2. The reaction with different nucleophiles.

- Abstract line 1: “The molecular...” => “Molecular...”

Response: Thank you for your kind suggestion. This error has been revised. Thanks again!

- Abstract, line 7: “derivates” => “derivatives”

Response: Thank you for your kind suggestion. This error has been revised. Thanks again!

- Abstract, line 8: “this innovative method” => “this method”

Response: Thank you for your kind suggestion. This error has been revised. Thanks again!

- Abstract, line 11: “synthetic application” => “synthetic applications”

Response: Thank you for your kind suggestion. This error has been revised. Thanks again!

- Introduction, line 9: “could precise swap” => “could precisely swap”

Response: Thank you for your kind suggestion. This error has been revised. Thanks again!

- Studies of synthetic applications, line 12: “...and yields higher with fewer operation steps.” => “and provide higher yields with fewer operation steps.”

Response: Thank you for your kind suggestion. This error has been revised. Thanks again!

- Reaction mechanism study, line 8: “a cross-reaction experiment” => “a crossover experiment”

Response: Thank you for your kind suggestion. This error has been revised. Thanks again!

- SI pages S40-41. These products are quinolines, not quinolones.

Response: Thank you for your kind suggestion. This error has been revised. Thanks again!

REVIEWERS' COMMENTS

Reviewer #2 (Remarks to the Author):

We are satisfied with the latest revisions. We recommend the publication of this manuscript.